# CNNSAT: Fast, Accurate Boolean Satisfiability using Convolutional Neural Networks

## Abstract

Boolean satisfiability (SAT) is one of the most well-known NP-complete problems and has been extensively studied. State-of-the-art solvers exist and have found a wide range of applications. However, they still do not scale well to formulas with hundreds of variables for uniform 3-SAT problems. To tackle this fundamental scalability challenge, we introduce CNNSAT, a fast and accurate statistical decision procedure for SAT based on convolutional neural networks. CNNSAT's effectiveness is due to a *precise* and *compact representation* of Boolean formulas. On both real and synthetic formulas, CNNSAT is *highly accurate* and *orders of magnitude faster* than the state-of-the-art solver Z3. We also describe how to extend CNNSAT to predict satisfying assignments when it predicts a formula to be satisfiable.

## 1 Introduction

The Boolean satisfiability problem, or SAT, is a classical decision problem. Given a propositional formula $\phi$, SAT needs to decide whether $\phi$ has a satisfying assignment to its variables. If the answer is yes, we say that the formula $\phi$ is satisfiable, or SAT for short. Otherwise, it is unsatisfiable, or UNSAT for short. For example, the formula $(x_1 \vee x_2) \wedge (\neg x_1 \vee x_2) \wedge (x_1 \vee \neg x_2)$ is satisfiable when both $x_1$ and $x_2$ are true (*i.e.*, $x_1 = x_2 = \textbf{true}$). Conversely, $(x_1 \vee x_2) \wedge (\neg x_1 \vee x_2) \wedge (x_1 \vee \neg x_2) \wedge (\neg x_1 \vee \neg x_2)$ cannot be satisfied by any of the possible assignments.

In other words, SAT asks whether the variables of the given Boolean formula $\phi$ can be consistently assigned the values **true** or **false** such that the formula evaluates to **true**. If this is the case, the formula is called satisfiable. On the other hand, if no such assignment exists, the formula is **false** for all possible variable assignments and is unsatisfiable.

SAT is a classical NP-complete problem, and in fact was the first problem proved NP-complete. Many hard problems naturally reduce to SAT, such as the traveling salesman problem (TSP) and clique detection. SAT has been extensively studied in the literature for decades because it is a foundational problem and has wide applications. The more general satisfiability modulo theories (SMT) can also be reduced to SAT solving.

SAT is one of the most investigated problems, and numerous heuristics exist to help speed up SAT solving. However, state-of-the-art solvers do not yet scale to large, difficult formulas, such as ones with hundreds of variables and thousands of clauses for uniform 3-SAT problems (Xu et al., 2012). This is because the search space for solutions increases exponentially *w.r.t.* the number of variables. Most search-based SAT solvers are based on the DPLL approach (Davis et al., 1962), but the search space, even reduced, is still intractable for very large formulas.

State-of-the-art methods for SAT adopt the Conflict-Driven Clause Learning (CDCL) (Silva & Sakallah, 1996; Bayardo Jr & Schrag, 1997) algorithm. This is a systematic search algorithm but employs various optimizations to improve efficiency. However, because the general problem is NP-complete, systematic search algorithms have exponential worst-case complexity, which limits the scalability of these methods.

There exists several pieces of previous work that try to use machine/deep learning methods to improve SAT solvers (Loreggia et al., 2016; Fei & Rompf, 2018), classify SAT/UNSAT (Bünz & Lamm, 2017; Grozea & Popescu, 2014; Devlin & O'Sullivan, 2008), or directly solve SAT instances (Selsam et al., 2018). However, all these approaches focus on SAT problems with a small number of variables.

In this paper, we introduce CNNSAT, a fast and accurate technique based on Convolutional Neural Networks (Krizhevsky et al., 2012) to predict both satisfiability and satisfying assignments. Evaluated on a data set containing 3-SAT problems with up to 410 variables, CNNSAT is able to predict SAT/UNSAT with more than 95% accuracy and orders of magnitude faster than state-of-the-art solvers. We also introduce optimizations to further improve CNNSAT's scalability. As for the more general SMT problems, CNNSAT is able to predict their satisfiability with more than 73% accuracy.

## 2 PRELIMINARIES

**Convolutional neural network (CNN).** CNN is a class of deep, feed-forward artificial neural networks. Many successful applications of CNNs exist in image and sound processing. A CNN has an input and an output layer, as well as multiple hidden layers, which typically consist of convolutional layers, pooling layers, fully connected layers and normalization layers.

**The SAT and 3-SAT problems.** In Boolean logic, a formula is in conjunctive normal form (CNF) if it is a conjunction of one or more sub-formulas. For a CNF formula, each of its sub-formulas is called a clause, which is a disjunction of literals (*i.e.*, variables or their negations). The *clause-to-variable ratio* of a CNF formula is defined as the ratio of the number of clauses over the number of variables.

Each SAT instance can be represented in CNF. A CNF formula has a satisfying assignment iff there exists at least one assignment for each variable in the formula such that the formula evaluates to **true**. The objective of SAT solving is to determine whether or not a given formula is satisfiable, and produce a satisfying assignment when the formula is satisfiable.

3-SAT is a special case of SAT where the number of literals in each clause is up to three. Generalizing 3-SAT, $N$-SAT requires all clauses having no more than $N$ literals, and *uniform $N$-SAT* requires all clauses having *exactly* $N$ literals. 3-SAT is also NP-complete, and, in general, $N$-SAT, for $N > 2$, can be reduced to 3-SAT.

**The SMT problem** Satisfiability Modulo Theories (SMT) refers to the problem of determining whether a first-order formula is satisfiable *w.r.t.* some logical theories. It is typically applied to the theory of real numbers, the theory of integers, and the theories of various data structures, such as lists, arrays, and bit vectors.

For brevity, hereafter if a SAT problem instance is in CNF, we refer to it as CNF. Otherwise, we still use SAT to refer to the more general case.

## 3 CNNSAT

This section presents the technical details behind our approach. In particular, it describes the representation that we introduce to encode CNF formulas, the architecture of our proposed neural network, and the method that we use to find satisfying assignments.

### 3.1 REPRESENTATION

A SAT problem has a simple syntactic structure and therefore can be encoded into a syntax-based representation such as an abstract syntax tree (AST). The semantics of propositional logic induces rich invariance that such syntactic representations would ignore, *e.g.*, permutation and negation invariance (Selsam et al., 2018). Permutation invariance stipulates that the satisfiability of a SAT problem is not affected by swapping the variables (*e.g.*, swapping all occurrences of $x_1$ with those of $x_2$ in the SAT instance). Negation invariance means that negating every literal corresponding to a given variable (*e.g.*, replacing $x_i$ by $\neg x_i$, and $\neg x_i$ by $x_i$ for any variable $x_i$ in the SAT instance).

As noted by Selsam et al. (2018), syntax-based representations do not capture the semantics of SAT problems. In other words, they cannot identify even the simplest semantic equivalence among SAT problems, such as permutation and negation invariance discussed earlier. On the other hand, even though syntax-based representations may not accurately capture semantic equivalence, sufficient amount of training data may allow neural networks to learn and predict the semantics of SAT formulas. Our evaluation in Section 5 confirms this hypothesis. In addition, for certain applications, most CNFs do not share the same/similar semantics. Therefore, we adopt a syntax-based representation to balance accuracy and scalability.

```
p  CNF  5  6
1    2   3
2   -3   4
1   -2  -3
1    2   4
3   -4  -5
-3    4   5
```

| 1 | 1 | 1 | 0 | 0 |
|---|---|---|---|---|
| 0 | 1 | -1 | 1 | 0 |
| 1 | -1 | -1 | 0 | 0 |
| 1 | 1 | 0 | 1 | 0 |
| 0 | 0 | 1 | -1 | -1 |
| 0 | 0 | -1 | 1 | 1 |

Figure 1: Example to illustrate our representation.

**SAT Representation.** Although a SAT problem can be represented in different forms, we choose the most common CNF format. Each clause in a CNF formula $\phi$ is represented by a vector $v$, where $v = \langle e_1, e_2, ..., e_n \rangle$, and the dimension of $v$, $n$, corresponds to the number of variables in $\phi$. For each element $e_i$ in the vector, we set it to 0 if the corresponding variable $x_i$ does not occur in the clause, -1 if the variable $x_i$ is negated, and 1 otherwise (*i.e.*, when the literal $x_i$ appears in the clause). Collectively, the vectors for $\phi$'s clauses form an $m \times n$ matrix, where $m$ is the number of clauses and $n$ the number of variables.

Figure 1 shows an example to illustrate this representation. The CNF formula is shown in the left sub-figure, while the representation is shown in the right sub-figure. The first line in the left sub-figure (p CNF 5 6) indicates that the CNF has 5 variables and 6 clauses. The other rows in the left sub-figure is in the format $\langle v_{i1}\ v_{i2}\ v_{i3} \rangle$, where $i$ is the $i$-th clause, $v_{ij}$ is a literal (*i.e.*, $j$-th variable or its negation) in the clause — a negative value indicates that the variable is negated. The actual CNF formula is

$$(x_1 \vee x_2 \vee x_3) \wedge (x_2 \vee \neg x_3 \vee x_4) \wedge (x_1 \vee \neg x_2 \vee \neg x_3) \wedge (x_1 \vee x_2 \vee x_4) \wedge (x_3 \vee \neg x_4 \vee \neg x_5) \wedge (\neg x_3 \vee x_4 \vee x_5)$$

From the table, we can see that the representation in the right sub-figure encodes all the values of the variables into corresponding values in the matrix.

This representation is straightforward and the conversion is efficient. Note that this is a sparse matrix because only a small number of elements in each row are nonzero. However, we observe that, in practice, a SAT problem may have as many as millions of variables and clauses. At such a large scale, these SAT problems cannot fit in memory. Therefore, we propose a compact representation to improve scalability.

Our core idea is to split a matrix into smaller sub-matrices and summarize information for each sub-matrix. First, we define a fixed size sliding window. Then, we split the original matrix into sub-matrices according to the size of the original matrix and the sliding window. For each sub-matrix, $r_i = \langle p_i, n_i \rangle$ is a compact representation for the $i$-th sub-matrix, where $p_i$ is the number of positive values in the sub-matrix and $n_i$ the number of negative values. Therefore, each sub-matrix is converted to a list with 2 elements. It is worth noting that when the size of the sliding window is $1 \times 1$, it retains the exact information in the original matrix. In the next section, we introduce additional optimizations for the compact matrix for better performance. Our experimental evaluation shows that this representation can accurately capture semantic equivalence.

**SMT Representation.** There are several straightforward representations for SAT problems. In contrast, representing SMT problems is more challenging. Although we can design custom representations for SMT, we choose to translate SMT problems to SAT problems so that we can leverage our representation of SAT problems to also encode SMT problems.

## 3.2 Network Architecture

Figure 2 depicts the architecture of our proposed neural network which uses three convolution layers for CNN. The first layer of our network aims at reducing the scale of the input matrix because this matrix can still be too large to fit in memory even for the compact representation. The last two layers are used for building neural networks.

For convolution layers whose stride is one, the size of the output after one layer is sightly smaller than the size of the input. The output size depends on the kernel size. Therefore, the scalability of this model is poor if the size of the input is large. In order to tackle this challenge, we add the first

Figure 2: Network Architecture

---

**Algorithm 1:** Solving_CNF

---

**Input:** $\phi$, N
**Output:** Result

1  res := predictCNF($\phi$);
2  **if** *res = UNSAT* **then**
3  |   return UNSAT;

4  assignment := [];
5  predTimes := 0;
6  index := 0;
7  predLists := new map();
8  **while** *index < NumberOfVar($\phi$)* **do**
9  |   assign := random([**true**, **false**]);
10 |   newCNF := assignVar($\phi$, assign, index);
11 |   // res is a structure with $\langle label, probability \rangle$
12 |   res := predictCNF(newCNF);
13 |   predLists.insert(newCNF, res);

14 newCNF := chooseTopNProb($\phi$, predLists, N);
15 assignment := solver.solve(newCNF);
16 **if** *assignment = SAT* **then**
17 |   return contructAssign(assignment, predLists, N);

18 return UNKNOWN;

---

layer, whose goal is to shrink each input matrix into a fixed size matrix by choosing a specific stride and kernel size. At the high-level, we first split an input matrix into a fixed number of sub-matrices (*e.g.*, $100 \times 100$). *N and M are determined by the input matrix.* Then, we extract the features of each sub-matrix and use them to form a new matrix. In this way, we are able to process matrices of any size, and the only requirement is that the input matrix should pass the first layer.

After the first convolution layer, the size of the matrix is fixed (*e.g.*, $100 \times 100$). We then build three pooling layers and two other convolution layers. The last layer is a fully-connected layer that computes the scores.

### 3.3 SAT Solving

In order to solve a CNF formula instead of only predicting whether it is SAT or UNSAT, we simplify the CNF formula by guessing a satisfying assignment. We predict an assignment as follows. First, we construct new CNF formulas by assigning random values (*i.e.*, **true** or **false**) to variables, and thus construct new matrices. We then feed these new matrices to the trained model and analyze the prediction results. We choose a specific number of assignments based on prediction probabilities (*i.e.*, confidence). Next, we use an off-the-shelf solver to find assignments for the rest of the variables. Finally, we combine the two types of assignments to construct a final assignment.

Algorithm 1 shows the steps we use for solving CNF formulas. The input is a formula instead of a compressed matrix, which limits the scalability of satisfiability solving. First, we do not solve

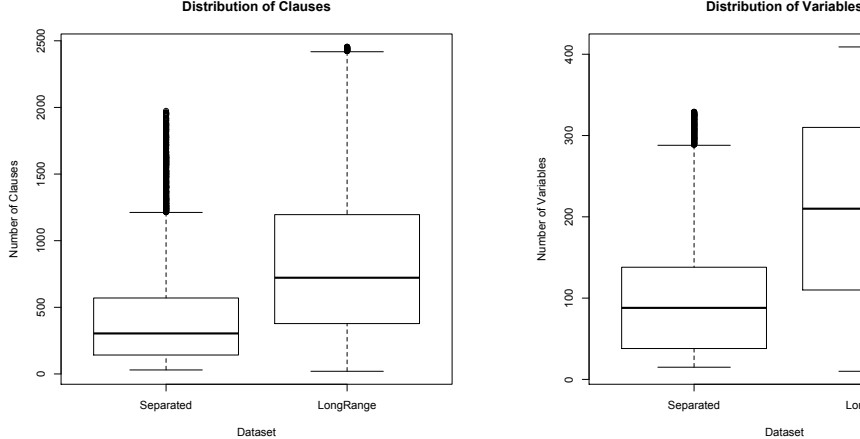

(a) Distribution of number of clauses in two dataset    (b) Distribution of number of variables in two dataset

Figure 3: Distribution of variables and clauses in each dataset

formulas that our CNN model predicts to be UNSAT (Lines 1-3). We assign random values (**true** or **false**) to the variables and use our model to predict them (Lines 10-12). Note that we assign variables one by one based on the order of variables (Line 8). Then, we store the result and the new CNF (Line 13). After obtaining prediction results for all variables, we select a specific number (*i.e.*, N) of predicted variables ranked by probability (Line 14). Reducing the original CNF formula with these partial assignments yields a new, simplified CNF formula, which is fed to an existing solver (Line 15). At the end, we merge the predicted partial assignment with the solver result to construct an assignment if the solver finds a satisfiable assignment (Lines 16-17). Otherwise we regard the formula as UNKNOWN (Line 18).

Consider, for example, an input CNF formula $(x_1 \vee x_2) \wedge (\neg x_1 \vee x_2) \wedge (x_1 \vee \neg x_2)$. First, assume that we assign **false** to $x_1$, which leads to the new, simplified CNF formula: $(x_2) \wedge (\neg x_2)$. We feed this formula to our model, and let us assume that it predicts the formula to be SAT with 80% probability. Next, we try $x_2$ as **true**, the CNF formula simplifies to $(x_1)$. If the prediction is SAT with 90% probability and the $N$ is 1, then we assign **true** to $x_2$ and use a solver to resolve $(x_1)$. The solver returns the satisfying assignment that $x_1 = $ **true**. With these two pieces of variable assignment information, we derive the satisfying assignment $\{x_1 = $ **true**$, x_2 = $ **true**$\}$ for the original CNF formula. Note that if $N$ were chosen to be 2, the combined variable assignment is not a satisfying assignment. We choose to determine $N$ dynamically based on the dataset.

## 4 DATASETS

We use CNFgen (Lauria et al., 2017) to generate CNF formulas in the DIMACS format. It generates combinatorial, challenging problems for SAT solvers. CNFgen is also able to generate different problems. For this work, we restricted CNFgen to generate random 3-SAT instances whose number of variables and number of clauses are configurable.

We generate two kinds of datasets, *Long Range* and *Separated*. The number of variables for *Long Range* ranges from 10 to 410 and the clause-variable ratio ranges from 4 to 8. It takes longer time for solvers to solve CNFs with more than 400 variables and 8 clause-variable ratio. We generate $16,000$ random CNFs.

The second dataset *Separated* is used to test the ability of CNNSAT when predicting CNFs with three smaller datasets. The data set consists of three sub-datasets: (1) a small dataset whose number of variables ranges from 12 to 30, (2) a medium dataset whose number of variables ranges from 130 to 160, and (3) a large dataset whose number of variables is between 300 and 330. The clause-variable ratio still ranges from 4 to 8. There are $95,000$ CNF formulas in this dataset.

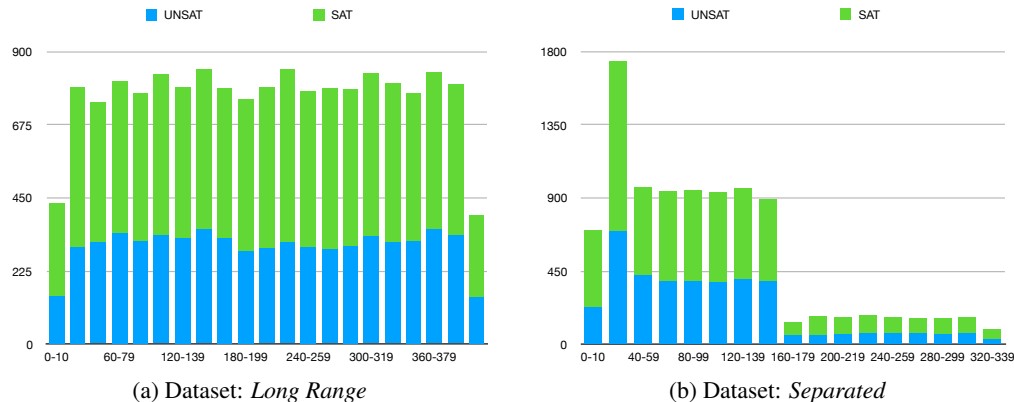

(a) Dataset: *Long Range*    (b) Dataset: *Separated*

Figure 4: Distribution on the number of variables in two datasets.

We use 75% of the whole dataset for training and the rest of them for testing. A dataset should contain a relatively balanced distribution of satisfiable and unsatisfiable instances, and cannot be made from instances that are all in the same class. The ratio of SAT to UNSAT is 9637:6357 in *Long Range* and the ratio of SAT to UNSAT is 5604:3896 in *Separated*.

Figure 3a depicts the number of clauses in the different datasets. Figure 3b shows the distribution of the number of variables in the different datasets. *Long Range* is a dataset that is unbiased *w.r.t.* the number of variables, but *Separated* is not. The goal of the *Separated* dataset is to compare the behavior of networks with balanced and unbalanced datasets.

Figures 4a and 4b show the distribution of SAT and UNSAT instances in the different datasets. The number of SAT and UNSAT instances in these datasets is nearly evenly distributed across different ranges of variables. Note that the number of variables is not evenly distributed (Figure 4b) because we would also like to evaluate the performance of CNNSAT when the dataset is not evenly distributed by the number of variables.

Finally, we construct our SMT dataset from the SMT benchmarks provided by SMT (2018). We choose two theories: QF_BV and QF_IDL. As for predicting satisfiability for SMT problems, we use Z3 (De Moura & Bjørner, 2008) to convert them to SAT problems and use our model to predict satisfiability for these SAT problems.

## 5    EVALUATION

All our experiments run on a PC with the following hardware configuration: Intel(R) Core(TM) i7-7700 CPU @ 3.60GHz, 16GB memory and the GPU is GeForce 730 with 2GB memory. We have implemented CNNSAT based on TensorFlow with GPU support.

As discussed earlier, we use CNFgen (Lauria et al., 2017) to generate random 3-SAT problem instances in the DIMACS format. We use Z3 (De Moura & Bjørner, 2008) to convert SMT problems to SAT problems. PicoSAT (Biere, 2007) is used to help predict assignments for CNF formulas. We discard all SAT problems that cannot be solved by PicoSAT within a 10-minute budget. For each dataset, 75% of the data is used for training and the rest for testing.

### 5.1    PREDICTION RESULTS ON RANDOM 3-SAT PROBLEMS

Table 1 shows the summary results of our neural network on different datasets. We evaluated CNNSAT's accuracy over the datasets with the 25% holdout setting, *i.e.*, we trained our models on 75% of the data and tested on the remaining 25% data. We performed all experiments three times and computed the average performance over these three runs.

Table 1 shows CNNSAT's accuracy on two datasets. The overall accuracy on the *Long Range* 3-SAT instances is 98.1%. The accuracy for SAT on SAT instances is 99.0%, and the accuracy for UNSAT on UNSAT instances is 97.0%. The accuracy for predicting satisfying assignments is 92.6%. The

Table 1: Accuracy of the Trained Model

| Dataset | Accuracy | | | | Time (Second) | | | % of Imp on assign |
|---------|----------|--------|----------|-----------|------|----------|----------|---------|
| | Overall | On sat | On unsat | On assign | Pred | Mini SAT | Pico SAT | |
| *Long Range* | 98.1% | 99.0% | 97.0% | 92.6% | 227 | 39109 | 39919 | 22.4% |
| *Separated* | 96.4% | 97.7% | 94.3% | 91.4% | 105 | 429 | 355 | -79.3% |

Table 2: Accuracy on permutation and negation

| Dataset | Operation | Accuracy | | | % of difference |
|---------|-----------|----------|--------|----------|-----------------|
| | | Overall | On sat | On unsat | |
| *Long Range* | permutation | 97.8% | 99.4% | 95.4% | 0.31% |
| *Long Range* | negation | 98.0% | 98.5% | 97.2% | 0.43% |
| *Separated* | permutation | 96.1% | 97.2% | 94.5% | 0.49% |
| *Separated* | negation | 96.2% | 97.6% | 94.2% | 0.24% |

overall accuracy for the *Separated* 3-SAT instances is 96.4%. The accuracy for SAT on the SAT instances is 97.7%, while the accuracy for UNSAT on the UNSAT instances is 94.3%. CNNSAT's accuracy for predicting satisfying assignments is 91.4%.

As for the scalability of CNNSAT, we evaluated it from three aspects. First, we measure the time spent on predicting the satisfiability of CNF formulas. We use Z3, PicoSAT, MiniSAT (Sorensson & Een, 2005), Glucose (Audemard & Simon, 2009), Dimetheus (Gableske, 2013) and CaDiCaL (Biere, 2017) for comparison to evaluate CNNSAT's efficiency. Due to space constraints, we only show the results for the two best performing solvers, MiniSAT and PicoSAT. The "Pred" means the time used when making predictions on the test data. Note that 1/4 of the CNF formulas were used for testing. "MiniSAT" and "PicoSAT" show the time that MiniSAT and PicoSAT spent on solving all the CNF formulas, respectively. The results show that CNNSAT clearly outperforms MiniSAT and "PicoSAT" by 1-2 orders of magnitude, making it practical for real-world use. "% of Imp on assign" denotes the percentage of improvement for our SAT solving algorithm compared to directly solving CNFs predicted as satisfiable using PicoSAT. We can observe that predicting speed for *Long Range* is improved when using our method. However, the performance for dataset *Separated* is decreased. The reason is that *Separated* contains less complicated CNFs and thus there is little improvement when CNNSAT could predict values for a part of the variables. In contrast, CNNSAT introduces additional overhead by predicting potential assignments.

## 5.2 EQUIVALENCE RESULTS

In this experiment, we evaluate two kinds of semantic equivalent operations, permutation invariance and negation invariance. For negation invariance, we generate datasets by negating half the variables. As for permutation invariance, we randomly choose two variables and swap them. For each CNF instance, we swap variables $\lfloor N/2 \rfloor$ times, where $N$ is the number of variables. For the two operations, we evaluate them three times and average the results.

Table 2 shows the results. We can see that CNNSAT predicts SAT/UNSAT with high accuracy. The corresponding accuracy is close to the original dataset in Table 1. The % of difference shows the percent of differences in individual predictions. The evaluation results show that CNNSAT is able to capture the semantics of SAT problems.

## 5.3 ACCURACY ON SMT BENCHMARKS

Table 3 shows the accuracy of CNNSAT on SMT benchmarks. The timeout for each phase is also 10 minutes. "CNV time" stands for how much time it takes to convert SMT problems to SAT problems. In our experiment, Z3 may convert an SMT to an empty SAT whose number of variable is zero or one. We ignore these trivial SAT instances.

We can see from the table that CNNSAT is able to predict them with more than 73% accuracy. In addition, CNNSAT is 1-2 orders faster than Z3.

Table 3: Accuracy SMT benchmarks

| Dataset | Accuracy | | | Time (Second) | | |
|---------|----------|--------|----------|----------|-----------|---------|
| | Overall | On sat | On unsat | CNV time | Pred time | Z3 time |
| QF_BV | 73.8% | 87.9% | 48.5% | 28,010 | 6,365 | 396,107 |
| QF_IDL | 91.2% | 97.5% | 44.6% | 701 | 157 | 138,656 |

## 5.4 DISCUSSIONS

**Sparse Convolutional Neural Network.** We use traditional CNN for CNNSAT, and construct a matrix based on CNF. However, it is clear that the matrix is sparse. In fact, for 3-SAT problems, the matrices are very sparse and most elements in these matrices are zero. However, we have not found sparse CNNs that best fit our scenario. Graham & van der Maaten (2017) present the Submanifold Sparse Convolutional Networks but since the matrices in our setting is not submanifold, it does not fit our representation.

**Guiding SAT solvers.** Most state-of-the-art SAT solvers implement Conflict-Driven Clause Learning (CDCL) (Silva & Sakallah, 1996; Bayardo Jr & Schrag, 1997). In CDCL, it continues selecting a variable and assigning **true** or **false**, and try to find conflict until all variable values are assigned. CNNSAT may improve its performance by trying to assign a variable the value leading the formula to SAT. Although the performance is not improved when a formula is UNSAT, it may improve performance when a formula is SAT. The performance can also be improved by learning the strategy that guiding the selection to choose a conflicting assignment.

## 6 RELATED WORK

Bello et al. (2017) present a framework to tackle combinatorial optimization problems using neural networks and reinforcement learning. They also apply it to other NP-hard problems such as traveling salesman problem and KnapSack. It shows performance improvement compared to standard algorithmic methods.

Fei & Rompf (2018) propose another avenue for SAT. They cast symbolic reasoning problems directly as gameplay to leverage the full decision-making power of neural networks through deep reinforcement learning. Most SAT solvers are based on the Conflict Driven Clause Learning (CDCL) algorithm, which is a typical symbolic reasoning process that can be cast as a game of controlling the branching decisions. The results show that this method can obtain better performance.

Xu et al. (2012) show that 70% classification accuracy can be obtained based on phase transition features on uniform-random 3-SAT formulas. CNNSAT's prediction accuracy is significantly higher under a similar experimental setup. In addition, phase transition features vary on different kinds of formulas, and thus a significant performance drop is expected on SAT instances converted from SMT formulas.

NeuroSAT (Selsam et al., 2018) uses an undirected graph to represent CNFs and builds a model by two vectors, three multilayer perceptrons and two layer-norm LSTMs. However, it needs to generate certain type of pairs to model SAT. In each pair, one element is satisfiable, the other is unsatisfiable, and the two differ by negating only a single literal occurrence in a single clause. Therefore, the training data is constrained by this requirement, which means for some data like uniform 3-SAT, it takes significant amount of time to generate the training data. In contrast, for CNNSAT, any training data is useful. NeuroSAT is unable to precisely predict satisfiability when the number of variables is large. Bünz & Lamm (2017) propose a method based on Graph Neural Network that is able to classify SATs with around 60% validation error. The representation is similar to NeuroSAT, which uses graphs to represent CNFs.

Feature-based machine learning methods Devlin & O'Sullivan (2008); Grozea & Popescu (2014) can also classify SATs. Grozea & Popescu (2014) aim to empirically test the ability of machine learning models to act as decision oracles for NP problems. They only evaluated the idea on formulas with up to 100 variables. The approach does not scale to formulas with more variables, such as those large formulas considered in this paper. Devlin & O'Sullivan (2008) view the satisfiability problem as a classification task. Based on easy to compute structural features of instances of large satisfiability

problems, they use a variety of standard classifier learners to classify previously unseen instances of the satisfiability problem as either SAT or UNSAT. The accuracy for classification is more than 90%. In comparison, CNNSAT can predict variable assignments and handle much larger formulas.

## 7 CONCLUSION

In this paper, we have introduced a new fast and accurate approach for solving SAT problems via Convolutional Neural Networks. We have described how we represent SAT instances, how we design our proposed neural network, how we optimize our technique for scalability, and our extensive evaluation to show CNNSAT's high accuracy and scalability on large SAT and SMT problem instances. Because CNNSAT's effectiveness, it may find interesting applications in domains that require fast SAT and SMT solving, such as software analysis and verification, symbolic execution, planning and scheduling, and combinatorial design.

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
