# OpenReview forum: "CNNSAT: Fast, Accurate Boolean Satisfiability using Convolutional Neural Networks"
_ICLR.cc/2019/Conference_

### Official Review · AnonReviewer2 · 2018-10-31
**Promising neural SAT solver, though limited contributions**

**Rating:** 5
**Confidence:** 4

**Review:**

The aim of this paper is to solve SAT instances using a CNN architecture. SAT instances are represented using an efficient encoding of boolean matrices. The overall idea is to decompose an input SAT instance into simpler ones, and to train the neural model on simpler instances using an existing solver for labeling these instances. Based on satisfaction probabilities induced from simpler formulas, the architecture predicts a partial assignment which is fed to the existing solver for deriving the satisfiability result.

Arguably, the topic of “learning to solve SAT instances” is very interesting, by coupling results from neural networks and SAT solvers. This work is inspired from the landmark paper on NeuroSAT, and the experimental results look promising.

However, since the framework is focused on solving random SAT problems (especially random 3-SAT instances), the paper is missing a detailed description of this active research topic in AI and the SAT community (see e.g. [1,2]). Notably, the problem of generating realistic random k-SAT instances has long been considered as one of the most important challenges in SAT research [3]. Importantly, modern random k-SAT instances are not only characterized by their number of variables, and their ratio  #clauses / #variables, but with an additional “structure” which mimics real-world, industrial instances (see e.g. [4]).

Furthermore, I had some trouble understanding how a SAT instance is solved using algorithm 1. Specifically the text in Section 3.3 that explains Algorithm 1 is a bit confusing. How do “we choose a specific number of assignments based on prediction probabilities”? Unless I missed something, the output of the CNN architecture is a probability value that the input formula is SAT, so I don’t really see how this can be related to prediction probabilities of assignments. This should be explained in detail since Line 15 is the main output of the algorithm, which is fed (Line 16) to an existing solver for completing the assignment. The example at the end of section 3.3 is not very helpful: namely, the CNF formula $(x_2) \land (\neg x_2)$ is clearly unsatisfiable, so how can the model predict that it is satisfiable with 80% probability? And, if we try here $x_2 = 1$, we immediately get $\bot$ (the unsat CNF), but not $x_1$ (which was already assigned to $0$).

Finally, the CNN architecture should be compared with modern SAT solvers which have been participating to SAT competitions. The Z3 solver is mainly focused on solving SMT instances [5], not random k-SAT instances which, by the way, is a common track in annual SAT competitions (see e.g. [6]). To this point, generic SAT solvers such as MiniSAT [7] and Glucose [8] are able to solve in few seconds some random 3-SAT instances with thousands of variables and tens of thousands of clauses (see e.g. [4]). So, the motivating assertion “[...] state-of-the-art solvers do not yet scale to large, difficult formulas, such as ones with hundreds of variables and thousands of clauses” in the introduction of the paper, is not totally correct. To sum up, I would recommend to compare the CNNSAT architecture with well-known SAT solvers such as MinSAT, Glucose, March, or Dimetheus [9] which has been one of the strongest solvers in recent years for tackling random instances. Also, as mentioned above, it would be interesting to incorporate some structures (such as, for example, community attachments or popularity-similarities) in SAT instances, in order to estimate whether CNNSAT could handle pseudo-industrial problems.

[1] D. Mitchell, B. Selman, H. Levesque, Hard and easy distributions of SAT problems, in: Proceedings of the 10th National Conference on Artificial Intelligence, AAAI’92, 1992, pp. 459–465.

[2] Nudelman, E., Leyton-Brown, K., Hoos, H. H., Devkar, A., & Shoham, Y. Understanding random SAT: Beyond the clauses-to-variables ratio. In 10th International Conference on Principles and Practice of Constraint Programming (CP’04), pp. 438–452.

[3] B. Selman, H.A. Kautz, D.A. McAllester, Ten challenges in propositional reasoning and search, in: Proceedings of the 15th International Joint Conference on Artificial Intelligence, IJCAI’97, 1997, pp. 50–54.

[4] J. Giráldez-Cru and J. Levy. Generating sat instances with community structure. Artificial Intelligence, 238:119 – 134, 2016.

[5] The 2014 SMT Competition https://satassociation.org/jsat/index.php/jsat/article/download/122/114

[6] The 2018 SAT Competition
http://sat2018.forsyte.tuwien.ac.at/index.php?cat=results

[7] N. Eén, N. Sörensson, An extensible SAT-solver, in: Proceedings of the 6th International Conference on Theory and Applications of Satisfiability Testing, SAT’03, 2003, pp. 502–518.

[8] ] G. Audemard, L. Simon, Predicting learnt clauses quality in modern SAT solvers, in: Proceedings of the 21st International Joint Conference on Artificial Intelligence, IJCAI’09, 2009, pp. 399–404

[9] Dimetheus
https://www.gableske.net/dimetheus

---

> ### Author Response · Authors · 2018-11-24
> **Response to AnonReviewer2**
>
> Thank you for the helpful comments and suggestions.
>
> Regarding the example, the goal of it is to help the reader understand
> the algorithm more easily. The wrong assignment with 80% probability
> is used to illustrate the situation where some predictions are wrong.
> "if we try here $x_2 = 1$" is based on the original formula, which is
> "$x_1 \lor x_2) \land (\lnot x_1 \lor x_2) \land ( x_1 \lor \lnot
> x_2)$". Therefore, the assigned formula is $x_1$.
>
> We tested PicoSAT, MiniSAT, Dimetheus and CaDiCaL and reported the
> results in the updated paper. CNNSAT outperformed all these solvers by
> at least two orders of magnitude over the "Long Range" dataset.

---

> > ### Comment · AnonReviewer2 · 2018-11-27
> > **Comments about Experiments**
> >
> > I have read the revised version of the paper, but I am still not convinced about the experimental setup and results. Notably:
> >
> > (i) I would suggest focusing on SAT generators or available benchmarks which have been used in SAT competitions. For example :
> >
> > - the generator which has been used in SAT Competitions since 2002 (this is, in particular, the generator used by Xu et. al., AAAI'12)
> >
> > https://www.satcompetition.org/2003/TOOLBOX/genAlea.c
> >
> > - or, preferably (for better reproducibility of the results), the random track benchmarks which have been used for SAT competitions:
> >
> > https://www.satcompetition.org/
> > http://sat2018.forsyte.tuwien.ac.at/benchmarks/
> >
> > By the way, the generator used in the last SAT competition is described here:
> >
> > https://helda.helsinki.fi/bitstream/handle/10138/237063/sc2018_proceedings.pdf?
> >
> > (ii) I don't understand why you "have discarded all SAT problems that cannot be solved by PicoSAT within a 10-minute budget." PicoSAT is arguably not the best solver for random problems, and 10mn is too short for tackling hard instances (close to the phase transition).
> >
> > So, again, it would be more relevant to use a standard setup. For example,
> > - take the random benchmarks used at the SAT'18 Competition:
> >
> > http://sat2018.forsyte.tuwien.ac.at/benchmarks/
> >
> > - Use only 3-sat instances
> >
> > - Compare CNNSAT with all standard solvers (Glucose, Dimetheus, etc.) using a timeout of 5000s (which is the standard value for several years in SAT Competitions), and a memory of 24 GB (which is again a standard parameter). Of course, CNNSAT can be trained on arbitrary random 3-SAT instances, but it should be tested on standard benchmarks, in order to highlight its performance against the state-of-the-art solvers.
> >
> > (iii) Finally, using the above benchmarks, it would be nice to display scatter plots for CNNSAT against other solvers. Don't hesitate to put them in the Appendix for space reasons, but they are much more informative than just a summarizing Table. Specifically, I am curious about the plot of CNNSAT vs Dimetheus (the last one is considered by the SAT community to be one of the best solvers for random instances).

---

> > > ### Author Response · Authors · 2018-11-27
> > > **Re "Comments about Experiments"**
> > >
> > > Thank you for your time and the additional comments.  We are happy to perform any additional experiments that the reviewer deems relevant, although we contend that we have demonstrated, via a set of extensive experiments, the orders of magnitude speedups of CNNSAT over the state-of-the-art solvers, including MiniSAT, PicoSAT, Dimetheus, etc.
> > >
> > > More information is also posted in our response to AnonReviewer1 and AnonReviewer3.  As it contains additional details that are not in the revised paper, could you please also look at the response if you have not already?   Thank you.
> > >
> > > Meanwhile, we will conduct additional experiments and post relevant results here.

---

> > > > ### Comment · AnonReviewer2 · 2018-11-28
> > > > **Re Re "Comments about Experiments"**
> > > >
> > > > Yes, sure, I’ve read all comments. But let’s try to summarize and, please, correct me if I’m wrong.
> > > >
> > > > The key message of the paper is that CNNSAT is in position to outperform the state-of-the-art solvers on random 3-SAT instances. Right?
> > > >
> > > > So, in order to prove this assertion, and to really highlight the performance of CNNSAT:
> > > > * The experimental setup should be as close as possible to the setup of the SAT competitions for the track on random instances.
> > > > * CNNSAT should be compared with the solvers which have won (or at least, which have been in good position to win) the competitions on random instances. Note that some of the winners are not CDCL solvers.
> > > > * Ideally, it would be nice to use scatter plots (or similar plots) for providing as much information as possible about the results.
> > > >
> > > > For now, in the revised version of the paper, we essentially have experimental results on two datasets. The datasets were generated from CNFGen (which is fine, I’m not criticizing this generator), but :
> > > > * All SAT problems that couldn’t be solved by PicoSAT within a 10-minute budget have been discarded;
> > > > * CNNSAT was trained on 75% of those instances, and tested on the remaining 25% instances, and
> > > > * For the results, we only have a two-row table about CNNSAT vs MiniSAT and PicoSAT, which are two CDCL solvers.
> > > >
> > > > Based on this protocol, we can hardly conclude that CNNSAT is outperforming most SAT solvers on random 3-SAT instances. We just can say something like  “For the 3-SAT instances (in which clauses are uniformly generated without replacement) that have been solved by PicoSAT within a range of 10 minutes, and using 75% instances for training, CNNSAT outperforms MiniSAT and PicoSAT on the remaining test instances.”
> > > >
> > > > Now, based on some of your comments (which are not included in the paper):
> > > >
> > > > > We evaluated the performance of CNNSAT on the Agile benchmarks from SAT competition
> > > > > 2016. The results are as follows.  The benchmarks took PicoSAT more than 48 hours to solve
> > > > > (the budget for each was 10  minutes).  CNNSAT solved all of them in 287 seconds with
> > > > > 97.5% accuracy --- the accuracy for UNSAT is 97.6%, while the accuracy for SAT is 97.1%.
> > > >
> > > >
> > > > For the sake of fairness, don’t use PicoSAT (which is not the best solver for random instances) for prunning out instances, and don’t use a 10mn timeout. I would suggest to use the recent winners here : Dimetheus or YalSAT, with a standard 5000s timeout. Ideally, CNNSAT should not be trained on the SAT’16 benchmark instances (which are obviously “test” instances); it should be trained on the instances of other SAT’<16 benchmarks , or using the generator for those instances (Tomas Balyo). Finally, the results should be reported in the paper.
> > > >
> > > > > Below are the results when considering the phase transition over the "Long Range" dataset.
> > > > > We follow Xu et al., 2012 and consider those instances with clause-to-variable (c/v) ratio
> > > > > in [3.26, 5.26], where a ratio of 4.26 is the solubility phase transition. Out of the 16,000
> > > > > instances in the "Long Range" dataset, 6,437 have c/v ratio within [3.26, 5.26].
> > > > > We use 4,827 for training and 1,610 for testing (803 of which are SAT instances
> > > > > and 807 are UNSAT instances), and obtain the following results:
> > > >
> > > > For analyzing the performance of CNNSAT at the phase transition, I would recommend to explicitly follow the protocol of Xu et al. 2012. Notably,
> > > > * Use the generator : https://www.satcompetition.org/2003/TOOLBOX/genAlea.c
> > > > * Use the solver kcnfs07 with a budget of 36,000s for labeling the instances.
> > > > And, again, the results should be reported in the paper, or in the Appendix.

---

> > > > > ### Author Response · Authors · 2018-11-28
> > > > > **Re Re Re "Comments about Experiments"**
> > > > >
> > > > > > Yes, sure, I’ve read all comments.
> > > > >
> > > > > Thank you.
> > > > >
> > > > > > The key message of the paper is that CNNSAT is in position to outperform the state-of-the-art
> > > > > > solvers on random 3-SAT instances. Right?
> > > > >
> > > > > Yes, that is accurate.   Again, we wish to contend that our experiments are already extensive and demonstrate CNNSAT's orders of magnitude performance gains over state-of-the-art solvers.
> > > > >
> > > > > To make the case even stronger and more convincing to the reviewer, we will sincerely follow the reviewer's suggestions to conduct further experiments and summarize our results.
> > > > >
> > > > > > So, in order to prove this assertion, and to really highlight the performance of CNNSAT:
> > > > > > ...
> > > > > > * Ideally, it would be nice to use scatter plots (or similar plots) for providing as much
> > > > > > information as possible about the results.
> > > > >
> > > > > This is a good suggestion; thank you, and we will.
> > > > >
> > > > > > For now, in the revised version of the paper, we essentially have experimental results on two
> > > > > > datasets. The datasets were generated from CNFGen (which is fine, I’m not criticizing this
> > > > > > generator), but :
> > > > > > * All SAT problems that couldn’t be solved by PicoSAT within a 10-minute budget have been
> > > > > > discarded;
> > > > > > * CNNSAT was trained on 75% of those instances, and tested on the remaining 25% instances, > and
> > > > > > * For the results, we only have a two-row table about CNNSAT vs MiniSAT and PicoSAT, which > are two CDCL solvers.
> > > > > >
> > > > > > Based on this protocol, we can hardly conclude that CNNSAT is outperforming most SAT
> > > > > > solvers on random 3-SAT instances. We just can say something like  “For the 3-SAT instances
> > > > > > (in which clauses are uniformly generated without replacement) that have been solved by
> > > > > > PicoSAT within a range of 10 minutes, and using 75% instances for training, CNNSAT
> > > > > > outperforms MiniSAT and PicoSAT on the remaining test instances.”
> > > > >
> > > > > We wish to make some clarifications:
> > > > >
> > > > > (1) PicoSAT was able to solve each formula in the "Separated" dataset within the 10-minute budget.  For the "Long Range" dataset, it could not solve only 6 formulas (out of the 16,000 formulas in the dataset) within the 10-minute budget.  Therefore, we chose this time budget because it seems to be sufficient for both datasets.  However, we are willing to solve the remaining 6 formulas using whatever resources (e.g., much larger time budgets) and redo our experiments.
> > > > >
> > > > > (2) We also compared the performance with other solvers besides MiniSAT and PicoSAT.   In particular, Section 5.1 mentions that "We use Z3, PicoSAT, MiniSAT, Glucose, Dimetheus, and CaDiCaL for comparison to evaluate CNNSAT's efficiency. Due to space constraints, we only show the results for the two best-performing solvers."   Thus, the results show that “For the 3-SAT instances (in which clauses are uniformly generated without replacement) that have been solved by PicoSAT within a range of 10 minutes, and using 75% instances for training, CNNSAT outperforms all six solvers (not only MiniSAT and PicoSAT) on the remaining test instances.”
> > > > >
> > > > > > Now, based on some of your comments (which are not included in the paper):
> > > > > > ...
> > > > > > For the sake of fairness, don’t use PicoSAT (which is not the best solver for random instances) > for prunning out instances, and don’t use a 10mn timeout. I would suggest to use the recent > winners here : Dimetheus or YalSAT, with a standard 5000s timeout. Ideally, CNNSAT should
> > > > > > not be trained on the SAT’16 benchmark instances (which are obviously “test” instances); it
> > > > > > should be trained on the instances of other SAT’<16 benchmarks , or using the generator for
> > > > > > those instances (Tomas Balyo).
> > > > >
> > > > > As commented earlier, we are happy to conduct any necessary experiments to make our results more convincing to the reviewer, so we will perform this experiment.
> > > > >
> > > > > Please do note that on the "Long Range" dataset generated by CNFGen, Dimetheus performed worse than both MiniSAT and PicoSAT, and CNNSAT outperformed both by two orders of magnitude (Section 5.1).
> > > > >
> > > > > > Finally, the results should be reported in the paper.
> > > > >
> > > > > Yes, we will (when we are permitted to revise the paper again).
> > > > > For now, we will share any results requested by the reviewer that we obtain on this discussion forum.
> > > > >
> > > > > > > Below are the results when considering the phase transition over the "Long Range" dataset.
> > > > > > > We follow Xu et al., 2012 and consider those instances with clause-to-variable (c/v) ratio
> > > > > > ...
> > > > > > For analyzing the performance of CNNSAT at the phase transition, I would recommend to
> > > > > > explicitly follow the protocol of Xu et al. 2012. Notably,
> > > > > > * Use the generator: https://www.satcompetition.org/2003/TOOLBOX/genAlea.c
> > > > > > * Use the solver kcnfs07 with a budget of 36,000s for labeling the instances.
> > > > >
> > > > > We will also do and report relevant results here as soon as we are able to obtain them.
> > > > >
> > > > > > And, again, the results should be reported in the paper, or in the Appendix.
> > > > >
> > > > > Yes, we will.

---

> > > > > ### Author Response · Authors · 2018-12-06
> > > > > **Part 1: Additional Experiments and Results**
> > > > >
> > > > > We apologize first for the delayed update because the additional
> > > > > experiments took significant amount of time. They are completed now,
> > > > > and below summarizes the main results.
> > > > >
> > > > > Note: All solvers were configured with 5000s budget in these experiments.
> > > > >
> > > > >
> > > > > --------
> > > > >
> > > > >
> > > > > (1) CNNSAT vs. state-of-the-art solvers on SAT competitions' random track instances
> > > > >
> > > > > We tried to perform the following experiment:
> > > > > - train CNNSAT on instances from the random track in 2016
> > > > > - test it on the instances from the random track in 2017
> > > > > - use 5000s for the timeout budget for each instance
> > > > > - compare CNNSAT with Dimetheus and YalSAT, the two best performing solvers for the random track
> > > > >
> > > > > Dimetheus was the winner for RandomSAT 2016 (with 5000s budget):
> > > > > - time: 41h53m with 5 threads
> > > > >
> > > > > YaleSAT was the winner for RandomSAT 2017 (with 5000s timeout):
> > > > > - time: 49h7m with 5 threads
> > > > >
> > > > > From our results from running Dimetheus and YaleSAT, and information
> > > > > on the SAT competition results, we did not find any UNSAT instances:
> > > > > - RandomSAT 2016: 128 (SAT) + 112 (UNKNOWN) = 240 (total)
> > > > > - RandomSAT 2017: 211 (SAT) + 89 (UNKNOWN) = 300 (total)
> > > > > - RandomSAT 2018: 243 (SAT) + 12 (UNKNOWN) = 255 (total)
> > > > > where we also attempted the RandomSAT 2018 instances.
> > > > >
> > > > > Thus, these are not good datasets for comparing CNNSAT and the other
> > > > > solvers because CNNSAT would have 100% accuracy (since there are not
> > > > > any UNSAT instances).
> > > > >
> > > > >
> > > > > --------
> > > > >
> > > > >
> > > > > (2) Use the "Long Range" dataset with 5000s budget, and compare with Dimetheus and YalSAT
> > > > >
> > > > > After more than 86 hours, Dimetheus and YalSAT only solved 155 and 147
> > > > > instances, respectively.  Please note that there are 16000 formulas in
> > > > > this dataset.  Even when using 10min budget (as we did and reported
> > > > > earlier), the two solvers did not perform nearly as well as MiniSAT
> > > > > and PicoSAT.
> > > > >
> > > > >
> > > > > --------
> > > > >
> > > > >
> > > > > (3) Experiment on the Agile benchmarks from the SAT competitions
> > > > >
> > > > > CaDiCaL and Riss: CaDiCaL Agile took 35h5m to solve the formulas in
> > > > > Agile 2017 with 5000s budget, 64G memory, and 10 solving instances. It
> > > > > would take much more time to solve the formulas one by one. CaDiCal
> > > > > took 75h to solve 913 formulas (out of the 5000 formulas in each of
> > > > > Agile 2016 and 2017).  It took the Riss solver more than 48h to solve
> > > > > the formulas in Agile 2016 with 5 solving instances.
> > > > >
> > > > > CNNSAT trained and tested on Agile 2016:
> > > > > - Accuracy: 99.20% overall (99.7% over UNSAT instances, 98.2% over SAT instances)
> > > > > - Prediction time: 522.72 seconds
> > > > >
> > > > > CNNSAT trained on Agile 2016 and tested on Agile 2017:
> > > > > *** [Case 1] without matched formula size (i.e., training on Agile 2016 and testing on all Agile 2017)
> > > > > - Accuracy: 84.9% overall (96.8% for UNSAT, 70.0% for SAT)
> > > > > - Prediction time: 17574 seconds
> > > > >
> > > > > *** [Case 2] with matched formula size (i.e., training on Agile 2016
> > > > > and testing on those Agile 2017 formulas with no more than the maximum
> > > > > number of variables for Agile 2016 formulas)
> > > > > - Accuracy: 99.1% overall (99.3% for UNSAT, 98.8% for SAT)
> > > > > - Prediction time: 1236 seconds
> > > > >
> > > > > Statistics on the Agile benchmarks:
> > > > > - Agile 2016: total 5000 formulas with 2207 SAT, 2777 UNSAT, and 16 timeout
> > > > > - Agile 2017: total 5000 formulas with 2187 SAT, 2744 UNSAT, and 69 timeout
> > > > > - Overlapping formulas in Agile 2016 and 2017: We found 3095 formulas
> > > > >   in Agile 2017 are the same as in Agile 2016.  Further inspecting the
> > > > >   datasets, we found that there were 19 new formulas in Agile 2017
> > > > >   with fewer than the maximum number of variables in Agile 2016.  If
> > > > >   we test on these 19 formulas, CNNSAT's overall accuracy is 94.7%
> > > > >   (100% for UNSAT, 94.4% for SAT).

---

> > > > > > ### Comment · AnonReviewer1 · 2018-12-06
> > > > > > **clarification question**
> > > > > >
> > > > > > You said:
> > > > > >
> > > > > > "From our results from running Dimetheus and YaleSAT, and information
> > > > > > on the SAT competition results, we did not find any UNSAT instances:
> > > > > > - RandomSAT 2016: 128 (SAT) + 112 (UNKNOWN) = 240 (total)
> > > > > > - RandomSAT 2017: 211 (SAT) + 89 (UNKNOWN) = 300 (total)
> > > > > > - RandomSAT 2018: 243 (SAT) + 12 (UNKNOWN) = 255 (total)
> > > > > > where we also attempted the RandomSAT 2018 instances."
> > > > > >
> > > > > > Are these numbers (e.g., 128 (SAT) + 112 (UNKNOWN) for the 2016 instances) the results of a single run of your method, with a time budget less than 5000s? Or what are they?
> > > > > >
> > > > > > If they are not the result of running your method, what would be the result of running your method? It's perfectly fine that there are no UNSAT instances; it is known that the instances in this track are satisfiable, but the problem is to actually find a satisfying assignment. So, for many of these instances does your method find a satisfying assignment?
> > > > > >
> > > > > > Thanks!

---

> > > > > > > ### Author Response · Authors · 2018-12-07
> > > > > > > **Re "clarification question"**
> > > > > > >
> > > > > > > Thank you for this follow-up question.
> > > > > > >
> > > > > > > > Are these numbers (e.g., 128 (SAT) + 112 (UNKNOWN) for the 2016
> > > > > > > > instances) the results of a single run of your method, with a time
> > > > > > > > budget less than 5000s?
> > > > > > >
> > > > > > > No.
> > > > > > >
> > > > > > > > Or what are they?
> > > > > > >
> > > > > > > These are results from running Dimetheus and YaleSAT.
> > > > > > >
> > > > > > > > If they are not the result of running your method, what would be the
> > > > > > > > result of running your method? It's perfectly fine that there are no
> > > > > > > > UNSAT instances; it is known that the instances in this track are
> > > > > > > > satisfiable, but the problem is to actually find a satisfying
> > > > > > > > assignment. So, for many of these instances does your method find a
> > > > > > > > satisfying assignment?
> > > > > > >
> > > > > > > We wish to clarify that CNNSAT's focus is on predicting SAT/UNSAT, not
> > > > > > > solving.  As we commented on another thread, CNNSAT does not replace,
> > > > > > > but complements modern solvers to quickly and accurately predict
> > > > > > > satisfiability.  One interesting example application of CNNSAT is for
> > > > > > > symbolic execution --- if a (complex) path constraint is quickly
> > > > > > > predicted to be UNSAT, no expensive solver call is necessary, and the
> > > > > > > corresponding path does not need to be explored.
> > > > > > >
> > > > > > > We did not evaluate our solving extension on these instances. If the
> > > > > > > reviewer believes that this is important, we are happy to conduct this
> > > > > > > experiment and report back the results, even though the experiment
> > > > > > > would take some time.

---

> > > > > ### Author Response · Authors · 2018-12-06
> > > > > **Part 2: Additional Experiments and Results**
> > > > >
> > > > >
> > > > > --------
> > > > >
> > > > >
> > > > > (4)
> > > > >
> > > > > > For analyzing the performance of CNNSAT at the phase transition, I would recommend to
> > > > > > explicitly follow the protocol of Xu et al. 2012. Notably,
> > > > > > * Use the generator: https://www.satcompetition.org/2003/TOOLBOX/genAlea.c
> > > > > > * Use the solver kcnfs07 with a budget of 36,000s for labeling the instances.
> > > > >
> > > > > We tried to regenerate formulas following the protocol of Xu et
> > > > > al. 2012. However, it takes a very long time to solve uniform 3-SAT
> > > > > problems with large numbers of variables. For example, it takes 48
> > > > > hours with 28 solving instances to solve only 159 formulas with around
> > > > > 350 variables.  To make this experiment feasible, we follow Xu et
> > > > > al.'s protocol with 300-400 variables. We used genAlea and kcnfs07 to
> > > > > construct a dataset with 1292 SAT and 1292 UNSAT instances. We use 75%
> > > > > for training and 25% for testing (both training and testing have an
> > > > > SAT:UNSAT ratio 1:1).  CNNSAT's overall accuracy is 61% (97.83% for
> > > > > UNSAT, and 24.17% for SAT). There are 646 formulas used for testing,
> > > > > and CNNSAT's prediction time is 28.8 seconds.  In comparison, it takes
> > > > > CaDiCaL 8 hours to solve only 76 formulas.  We could not find Xu et
> > > > > al.'s code and experimental data for a more direct evaluation.
> > > > >
> > > > >
> > > > > --------
> > > > >
> > > > >
> > > > > In summary, our additional experiments show that CNNSAT not only can
> > > > > classify random 3-SAT instances, but also classify other types of SAT
> > > > > (e.g., those problems converted from SMT benchmarks or from the Agile
> > > > > benchmarks).  CNNSAT can learn and classify SAT/UNSAT instances with
> > > > > high accuracy without the need to inspect patterns in specific types
> > > > > of problems.
> > > > >
> > > > > Thank you again for your time and consideration. We will include these
> > > > > results in our revision if space permits or in an extended version
> > > > > that will be shared on arXiv.

---

### Official Review · AnonReviewer3 · 2018-11-02
**Interesting idea, problematic evaluation**

**Rating:** 6
**Confidence:** 2

**Review:**

[Second Update] I still find the method proposed in this paper appealing, and think that it may have practical applications in addition to providing significant research contributions. A key question that was raised by the other two reviewers was whether the proposed approach was fairly evaluated against existing state-of-the-art solvers. The authors have responded to these concerns by adding clarifications and new baselines to their paper. However, based on the discussions to date, I feel that I am not sufficiently familiar with related work on SAT solvers to say whether the other reviewers' concerns have been fully addressed. If they have been, I'd strongly lean towards accepting the paper. As for the concerns from my original review: the transferability experiments reported in the author comments below are quite informative, and I'd encourage the authors to incorporate them into the paper (or an appendix if space is an issue). I'd also encourage the author to incorporate the full comparisons against Z3, PicoSAT, MiniSAT, Glucose, Dimetheus, and CaDiCaL from Section 5.1. (I've updated my rating for the paper from 5 to 6, and my confidence score from 3 to 2.)

[First Update] Based on the feedback of the other two reviewers, I believe that I was missing some important context about SAT solvers when I wrote my initial review. Reviewer 1 and Reviewer 2 both raised serious concerns about the types of SAT instances that were used to evaluate the experimental setup, as well as about the use of Z3 as a baseline for solving random SAT instances. (No author response was provided.) Given this additional information, I've lowered my score for the paper from an 8 to a 5. I do think that the approach is interesting, but have reservations about the experimental evaluation and the claims made by the current submission. (Note: As the paper authors point out in the comment below, this update was mistakenly submitted a few days before the end of the rebuttal period.)

[Original Title] Interesting idea, impressive results for a first paper

[Summary] The authors propose a method of using convolutional neural networks to determine whether large boolean formulas (containing hundreds of variables and thousands of clauses) are satisfiable. The resulting models are accurate (correctly distinguishing between satisfiable and unsatisfiable formulas more between 90% and 99% of the time, depending on the dataset) while taking 10x - 100x less time than an off-the-shelf solver (Z3), offering slightly better quality on some problems and slightly worse quality on others. In addition to determining whether formulas are satisfiable, the authors propose and evaluate a method for finding satisfying assignments. They also evaluate their system on SMT benchmarks, where it also shows 10x-100x speed-ups, albeit with somewhat lower accuracy (e.g., 73% - 92% accuracy; I couldn't find baselines for these experiments).

[Key Comments] Unless I'm missing something major, I'd prefer to accept this paper, since the problem appears novel and the experimental results seem very promising for a first paper on a new problem.

[Pro 1] The paper seems polished and well-written. I generally found it well-motivated and easy to follow.

[Pro 2] To the best of my knowledge, the problem domain (machine learning for satisfiability problems that are so large that they are difficult to solve using conventional methods) is both novel and well-motivated.

[Pro 3] Algorithms seem conceptually straightforward (but might be a bit challenging to implement in practice due to the large input size), and yield excellent results. The magnitude of speed-ups reported in the paper (10x - 100x) is large enough to be exciting from a research perspective, and also seems like it should be large enough to have significant practical applications.

[Pro 4] Results are evaluated on a variety of different boolean satisfiability and SMT problems.

[Con 1] To improve reproducibility, it would be helpful if the authors could provide more details about their model training setup. Figure 2 is a good start, but adding details about the layer sizes, types of pooling layers used, and the model training setup would help clarify the experiments.

[Con 2] It seems like a significant number of labeled training examples (i.e., examples that are already known to be satisfiable or unsatisfiable) are needed in order to train a neural network. This seems like it could present a bootstrapping problem for certain domains: it may be computationally expensive to generate ground-truth labels for training examples, but a significant number of labels are needed to train a good prediction model. I'd be very interested to see a study of how well trained models transfer across domains: how well do models trained on one domain (e.g., a synthetic problem where labeled training data is cheap to generate) transfer to a different domain (e.g., a real-world problem where training labels are expensive to compute)? However, this is a minor point for a first paper on a new problem, and I think the paper is interesting enough to merit acceptance without such an analysis.

---

> ### Author Response · Authors · 2018-11-24
> **Response to AnonReviewer3**
>
> Thank you for the valuable comments and suggestions. We apologize for
> not submitting our response and updated paper sooner as we had been
> performing additional experiments and analyses, and updating the
> paper.  We understand from an email from the PC chairs that the author
> response and revision deadline is Monday, November 26, 2018, which has
> not yet passed.
>
> Con 1: Thank you for the suggestion; we added details for Figure 2.
>
> Con 2: We tried two experiments:
>
> (1) Using the "Long Range" dataset for training and SAT instances
> converted from QF_BV for testing: we tested SAT instances with fewer
> than 500 variables since the maximum number of variables of the
> training set is 400. The accuracy is 69% --- accuracy on UNSAT is 23%
> and 82% on SAT.  The overall accuracy is 4% lower but the accuracy on
> UNSAT is significantly worse (compared to 48%).
>
> (2) Using Agile from the SAT competition 2016 benchmarks for training
> and SAT instances converted from QF_BV for testing: The accuracy is
> 38.8%, which is lower. We also trained on Agile and tested on the
> "Long Range" dataset. The accuracy is 39.7%.
>
> These results suggest that it is more effective to train the model for
> a specific domain as one would expect.

---

> > ### Comment · AnonReviewer3 · 2018-11-25
> > **Response deadline**
> >
> > You're right about the November 26th response deadline. I apologize for the confusion — I misinterpreted the reviewer instructions, and will update my review again after the end of the rebuttal period.

---

> > > ### Author Response · Authors · 2018-11-25
> > > **Re "Response deadline"**
> > >
> > > We greatly appreciate your consideration; thank you!

---

> ### Author Response · Authors · 2018-12-02
> **Re "Second Update"**
>
>
> > [Second Update] I still find the method proposed in this paper appealing, and think that it may have practical
> > applications in addition to providing significant research contributions. A key question that was raised by the other
> > two reviewers was whether the proposed approach was fairly evaluated against existing state-of-the-art solvers. The
> > authors have responded to these concerns by adding clarifications and new baselines to their paper. However, based
> > on the discussions to date, I feel that I am not sufficiently familiar with related work on SAT solvers to say whether the
> > other reviewers' concerns have been fully addressed. If they have been, I'd strongly lean towards accepting the paper.
>
> Thank you.   If possible, it would be very helpful to hear from the other two reviewers.
>
> We are also running the additional experiments that we promised in our latest response to AnonReviewer2.  These experiments are taking time, and we hope to have the results ready by tomorrow and then share with the reviewers immediately.
>
> > As for the concerns from my original review: the transferability experiments reported in the author comments below
> > are quite informative, and I'd encourage the authors to incorporate them into the paper (or an appendix if space is an
> > issue).
>
> Thank you; we will.
>
> > I'd also encourage the author to incorporate the full comparisons against Z3, PicoSAT, MiniSAT, Glucose,
> > Dimetheus, and CaDiCaL from Section 5.1.
>
> Yes, we will.
>
> > (I've updated my rating for the paper from 5 to 6, and my confidence score from 3 to 2.)
>
> Thank you for your time and consideration.

---

### Official Review · AnonReviewer1 · 2018-11-03
**Interesting problem & method, but misleading presentation**

**Rating:** 5
**Confidence:** 4

**Review:**

The authors present CNNSAT, a CNN-based approach to predict the satisfiability of SAT instances.
The problem is very relevant, and the approach is interesting, but unfortunately, the presentation is very misleading (see details below). In terms of methods, the main innovation appears to be the use of CNNs for predicting the solubility of SAT instances, but because of the exchangeability I don't actually see the intuition for this (see details below). Overall, because of these issues, I do not think this paper is ready for publication.

Misleading parts:
=================

1. Already in the abstract the authors make an utterly wrong statement about SAT solvers:
"State-of-the-art solvers exist and have found a wide range of applications.  However, they still do not scale well to formulas with hundreds of variables."
The same sentiment is repeated in the introduction; I'm puzzled why the authors would believe this. SAT solvers nowadays are routinely used on instances with hundreds of thousands of variables and millions of clauses (just see any of the recent SAT competitions (https://satcompetition.org/) for examples).

2. Z3 is *not* a good solver for random instances, far from state-of-the-art. It is not the right baseline.

3. The authors' approach implicitly makes use of PicoSAT, so their experiments are really implicitly comparing PicoSAT vs. Z3.

4. Comparing the prediction time of CNNSAT with the solving time of Z3 does not make much sense to me, since it does not solve every instance.

5. No comparison of CNNSAT (internally using PicoSAT) vs. PicoSAT is given.

6. The paper did *not* demonstrate that CNNSAT is competitive in practice. It is only faster than Z3 on random instances, which are not of practical interest. Demonstrating practical usefulness would require competitive performance on the SAT competition instances. By the way, CNNSAT would be disqualified in any SAT competition for falsely returning UNSAT for some satisfiable instances. Algorithm 1 should instead return UNKNOWN when it cannot find a solution.

7. Taken out of context, the predictive quality the authors achieve looks great: between 96% and 99% for random 3-SAT instances. However, this is misleading since the instances are not sampled at the phase transition and may thus be very easy to classify. Usually, for uniform random 3-SAT, the phase transition happens when the number of clauses for a number of variables v exceeds c = 4.258 * v +58.26 * v^{−2/3} (see [1]), although I do not remember whether this is for clauses being generated with and without replacement. (I looked into the documentation of CNFGEN, and for random k-cnf, it samples clauses without replacement.) It would be very useful to see the classification accuracy of classifying every formula with >= the number of clauses c from that formula as unsatisfiable and every formula with < that many clauses as satisfiable. Could the authors please report this number during the author response period?

The authors are also missing an additional related paper: [2] used simple models to obtain better-than-chance predictions at the phase transition.



Exchangeability and the use of CNNs:
====================================
Due to the exchangeability property, we do *not* care about spatial correlation in the adjacency matrix. I am really missing the details on how to achieve the fixed-size 100x100 matrices. This approach sounds like it would lose a lot of information!

I do not find the experiment studying exchangeability to be convincing. The experiment I would like to see is shuffling all variables, and/or negating half the variables, rather than swapping a single pair of variables. Even then, the experiment should optimally measure differences in individual predictions rather than differences in aggregate performance statistics.


One more question:
- How was N chosen? I only saw the statement "We choose to determine N dynamically based on the dataset."

[1] Crawford and Auton: Experimental results on the crossover point in random 3SAT. In Artificial Intelligence Journal, 1996.
[2] Xu, Hoos, and Leyton-Brown: Predicting Satisfiability at the Phase Transition. In AAAI 2012.

---

> ### Author Response · Authors · 2018-11-24
> **Response to AnonReviewer1**
>
> We appreciate the detailed, constructive feedback, which we discuss
> and respond to below.
>
> #1
>
> Thank you for pointing this out.  We changed the sentence to "they
> still do not scale well to formulas with hundreds of variables for
> uniform 3-SAT problems."
>
> We used MiniSAT and PicoSAT to solve the SAT instances in our dataset,
> all of which are uniform 3-SAT problems (see Figure 4 and Table 1):
> The "Long Range" instances take about 11 hours for both MiniSAT and
> PicoSAT to solve, while it only takes 429 seconds for MiniSAT and 355
> seconds for PicoSAT to solve those instances in "Separated".
>
> The main difference between the "Long Range" and "Separated" datasets
> is that "Long Range" contains more complex formulas with more than 300
> variables. Formulas in SAT competitions are not uniform 3-SAT
> problems.
>
> ------
>
> #2, #3, #5
>
> We evaluated CNNSAT against four SAT solvers, MiniSAT, PicoSAT,
> CaDiCaL and Dimetheus, and updated the paper with the additional
> results.  In summary, CNNSAT outperforms all of them by at least two
> orders of magnitude over the "Long Range" dataset.
>
> Our original submission had a typo in the sentence: "% of Imp on
> assign" denotes the percentage of improvement for our SAT solving
> algorithm compared to directly solving CNFs predicted as satisfiable
> using Z3.  Z3 should have been PicoSAT, which we have corrected in the
> updated paper.
>
> ------
>
> #6
>
> We evaluated the performance of CNNSAT on the Agile benchmarks from
> SAT competition 2016. The results are as follows.  The benchmarks took
> PicoSAT more than 48 hours to solve (the budget for each was 10
> minutes).  CNNSAT solved all of them in 287 seconds with 97.5%
> accuracy --- the accuracy for UNSAT is 97.6%, while the accuracy for
> SAT is 97.1%.
>
> ------
>
> #7
>
> Below are the results when considering the phase transition over the
> "Long Range" dataset.  We follow Xu et al., 2012 and consider those
> instances with clause-to-variable (c/v) ratio in [3.26, 5.26], where a
> ratio of 4.26 is the solubility phase transition. Out of the 16,000
> instances in the "Long Range" dataset, 6,437 have c/v ratio within
> [3.26, 5.26].  We use 4,827 for training and 1,610 for testing (803 of
> which are SAT instances and 807 are UNSAT instances), and obtain the
> following results:
>
> - CNNSAT's overall prediction accuracy: 95.5% (vs. 70% for Xu et
>   al., 2012)
>
> - CNNSAT's accuracy over the 803 SAT instances: 99.4%
>
> - CNNSAT's accuracy over the 807 UNSAT instances: 91.6%
>
> ------
>
> Re "Exchangeability and the use of CNNs"
>
> The goal of the exchangeability evaluation is to demonstrate that
> CNNSAT is able to identify semantic equivalent formulas.  Thank you
> for your suggestion; we have redone the experiment and updated the
> paper accordingly (see Table 2). In particular, we negated half of the
> variables and swapped half of the variables. There is only an average
> difference of 0.35% in prediction results from the results on the
> original formulas.
>
> The fixed-size 100x100 matrices are constructed by the first layer of
> CNN. We can define the size filter and stride based on the size of
> matrices. This compression is done by CNN.  Note that the matrix
> representation for CNF is very sparse, but the number of variables and
> clauses are very large (for SAT problems converted from SMT problems,
> the number of variables can be in the millions), which can easily lead
> to out-of-memory errors. Therefore, we compress the sparse matrices so
> that they are denser.
>
> In summary, we employ two types of compressions: (1) sparse matrix
> compression from the representation of CNF formulas, and (2)
> compression by the first layer of CNN.  The size of the compressed
> matrix (100x100) and N are both tuned with respect to accuracy or
> memory usage.

---

> > ### Comment · AnonReviewer1 · 2018-12-05
> > **Quick question**
> >
> > I'm just looking through the updates and have a quick question:
> > In my question 7, I asked: " It would be very useful to see the classification accuracy of classifying every formula with >= the number of clauses c from that formula as unsatisfiable and every formula with < that many clauses as satisfiable. Could the authors please report this number during the author response period?"
> >
> > The authors instead responded with a comparison to classification percentages from Xu et al. Are these exactly the same instances used by Xu et al? If not then I do not think that the percentages are comparable. Could the authors please answer my original question? Thanks!

---

> > > ### Author Response · Authors · 2018-12-06
> > > **Re: Quick question**
> > >
> > > Thank you for this follow-up question.
> > >
> > > > The authors instead responded with a comparison to classification
> > > > percentages from Xu et al. Are these exactly the same instances used
> > > > by Xu et al?
> > >
> > > These are not.  We could not find online the code or experimental data
> > > used by Xu et al.
> > >
> > > > If not then I do not think that the percentages are
> > > > comparable. Could the authors please answer my original question?
> > >
> > > (1) We did the calculation for the "Long Range" data set. Among the 1610
> > > testing instances, we have
> > > - 782 above phase transition point with	766 UNSAT and 16 SAT
> > > - 828 below the phase transition point with 787 SAT and 41 UNSAT
> > >
> > > The accuracy would have been (766+787)/1610 = 96.46%. Thus, the	simple
> > > classification performs very well on this dataset that mostly follows
> > > the phase transition.
> > >
> > > (2) On the Agile benchmarks from SAT competition 2016, all instances
> > > are below the phase transition point (with maximum c/v ratio 4.07).
> > > As commented under "#6", "CNNSAT solved all of them in 287 seconds
> > > with 97.5% accuracy --- the accuracy for UNSAT is 97.6%, while the
> > > accuracy for SAT is 97.1%."  Using the phase transition for
> > > classification, the accuracy would have been 50% as there are the same
> > > number of SAT and UNSAT instances for testing.

---

> > > > ### Comment · AnonReviewer1 · 2018-12-06
> > > > **update**
> > > >
> > > > I apologize for the delay in going through the revised version.
> > > > Having gone through the manuscript again, first, here are a few (easy to fix) remarks about the introduction:
> > > >
> > > > - The authors now cite Xu et al 2012, which is good, but they can't use this 6-year old paper as a reference for their controversial statement that state-of-the-art solvers do not scale to hundreds of variables (not even for uniform 3-SAT). This may be true for CDCL solvers, but I doubt it for stochastic local search solvers.
> > > >
> > > > - With the authors' new focus on random SAT, the intro is based on CDCL solvers too much; the entire paragraph on CDCL becomes irrelevant for random SAT. I would think the paper should either focus on random SAT (meaning a focus on random SAT in the intro, and in the experimental benchmarks and baselines) or  on real instances (meaning a focus on real problems in the intro, and in the experimental benchmarks).
> > > >
> > > > - I forgot to mention this earlier, but regarding previous work that tries to use machine learning methods to improve SAT solvers, there is additional related  work on this along two lines:
> > > > 1. algorithm selection (e.g., Xu et al, SATzilla: Portfolio-based Algorithm Selection for SAT, JAIR 2008), which won several SAT competitions by choosing the best algorithm for each SAT instance based on ML, and
> > > > 2. algorithm configuration (e.g., Hutter et al, Boosting Verification by Automatic Tuning of Decision Procedures, FMCAD 2007), which uses ML to tune SAT solver parameters.
> > > >
> > > >
> > > > A nitpick:
> > > > - N is used twice, once in the dimensions of the matrix (e.g., Figure 2), and once as the number of assignments to be made by the model.
> > > >
> > > > All of the above can be fixed easily.
> > > >
> > > >
> > > > The critical issue for me is whether the paper really demonstrates improvements over the state of the art in SAT solving. I am not yet convinced the authors really demonstrate that their approach outperforms modern SAT solvers (and their author response confirmed that this is supposed to be the paper's key message).
> > > > I believe we as the deep learning community have to be very careful with any sort of statement of this kind, because if it doesn't hold up this may lead to a backlash from the formal verification community against deep learning; we should thus be extra careful with statements of this kind and only make them when we can convincingly outperform their methods in their competition setting.
> > > >
> > > > Relatedly, the following sentence in the abstract appears inflated:
> > > > "On both real and synthetic formulas, CNNSAT is highly accurate and orders of magnitude faster than the state-of-the-art solver Z3".
> > > > -> apart from the fact that the sentence falsely identifies Z3 as a state-of-the-art solver for random instances (which would already cost the paper its credibility in the formal verification community), it hides the fact that the authors' approach actually makes mistakes, and that in practice no actual SAT solver would be allowed to make any mistakes.
> > > >
> > > > Incidentally, I also don't believe that the paper would have to demonstrate improvements over the state-of-the-art in SAT solving if the methods are interesting enough. Unfortunately, when trying to summarize to myself what the paper's novel methodological contribution is, I mainly could think of the use of convolutions, which I continue to find troublesome in this context. Due to exchangeability, there should be no local structure that can exploited. Also, I don't see any experiments where the authors study scalability of this approach (e.g., fraction of errors as a function of instance size). Finally, here is a conceptual question: if we always split instances into 10000 submatrices (100x100), and for each of these submatrices record the number of positive and negative literals, then couldn't an instance that has 1000 times more variables but the same number of literals end up with the same representation? That seems very suboptimal.
> > > >
> > > > Taking into account the authors' modifications I've increased my rating from 4 to 5, but because of the reasons above I hesitate to raise it higher.

---

> > > > > ### Author Response · Authors · 2018-12-07
> > > > > **Re "update"**
> > > > >
> > > > > Thank you.  We will make the small fixes that you mentioned.
> > > > >
> > > > > > The critical issue for me is whether the paper really demonstrates
> > > > > > improvements over the state of the art in SAT solving. I am not yet
> > > > > > convinced the authors really demonstrate that their approach
> > > > > > outperforms modern SAT solvers (and their author response confirmed
> > > > > > that this is supposed to be the paper's key message).
> > > > > > ...
> > > > >
> > > > > Our wording may be unclear, which we will state more accurately: (1)
> > > > > CNNSAT's focus is on predicting SAT/UNSAT, not solving, and (2) it
> > > > > achieves very accurate and fast SAT/UNSAT prediction.
> > > > >
> > > > > CNNSAT does not replace, but complements modern solvers to quickly and
> > > > > accurately predict satisfiability.  One interesting example
> > > > > application of CNNSAT is for symbolic execution --- if a (complex)
> > > > > path constraint is quickly predicted to be UNSAT, no expensive solver
> > > > > call is necessary, and the corresponding path does not need to be
> > > > > explored.
> > > > >
> > > > > We will paraphrase as follows to avoid misinterpretation: "On both
> > > > > real and synthetic formulas, CNNSAT is highly accurate and efficient
> > > > > in predicting satisfiability."
> > > > >
> > > > > > Incidentally, I also don't believe that the paper would have to
> > > > > > demonstrate improvements over the state-of-the-art in SAT solving if
> > > > > > the methods are interesting enough. Unfortunately, when trying to
> > > > > > summarize to myself what the paper's novel methodological
> > > > > > contribution is, I mainly could think of the use of convolutions,
> > > > > > which I continue to find troublesome in this context.
> > > > >
> > > > > We wish to note that the novel methodological contributions include
> > > > > (1) the use of convolutions and a novel compact representation to
> > > > > scale to formulas with large numbers of variables, and (2) a SAT
> > > > > solving algorithm based on satisfiability prediction to speed up
> > > > > constraint solving.
> > > > >
> > > > > > Due to exchangeability, there should be no local structure that can
> > > > > > exploited. Also, I don't see any experiments where the authors study
> > > > > > scalability of this approach (e.g., fraction of errors as a function
> > > > > > of instance size).
> > > > >
> > > > > Please note that our experiments show that CNNSAT can deal with large
> > > > > instances with excellent performance. Its time is rather invariant
> > > > > across different formula sizes. Below is some more information on the
> > > > > "Long Range" dataset to demonstrate this:
> > > > >
> > > > > (1) #var between 100 and 150:
> > > > > - total formulas: 522
> > > > > - prediction time: 53.0s (0.102s per formula)
> > > > > - accuracy: 99.23%
> > > > >
> > > > > (2) #var between 150 and 200:
> > > > > - total formulas: 503
> > > > > - prediction time: 52.4s ((0.104s per formula)
> > > > > - accuracy: 99.20%
> > > > >
> > > > > (3) #var between 200 and 250:
> > > > > - total formulas: 518
> > > > > - prediction time: 53.6s (0.104s per formula)
> > > > > - accuracy: 99.80%
> > > > >
> > > > > (4) #var between 250 and 300:
> > > > > - total formulas: 505
> > > > > - prediction time: 52.8s (0.105s per formula)
> > > > > - accuracy: 99.20%
> > > > >
> > > > > (5) #var between 300 and 350:
> > > > > - total formulas: 518
> > > > > - prediction time: 54.3s (0.105s per formula)
> > > > > - accuracy: 95.17%
> > > > >
> > > > > (6) #var between 350 and 400:
> > > > > - total formulas: 519
> > > > > - prediction time: 55.0s (0.106s per formula)
> > > > > - accuracy: 99.81%
> > > > >
> > > > > > Finally, here is a conceptual question: if we always split instances
> > > > > > into 10000 submatrices (100x100), and for each of these submatrices
> > > > > > record the number of positive and negative literals, then couldn't
> > > > > > an instance that has 1000 times more variables but the same number
> > > > > > of literals end up with the same representation? That seems very
> > > > > > suboptimal.
> > > > >
> > > > > Please note that such two instances would have very different initial
> > > > > representations because of their drastically different sizes, C1xV1
> > > > > vs.  C2xV2 (where C1, C2 the numbers of clauses, and V1, V2 the
> > > > > numbers of variables in the two instances): C1 = C2, but V1 <<
> > > > > V2. Thus, CNNSAT works well with similarly sized instances.
> > > > >
> > > > > CNNSAT's compressed representation can indeed lead to collisions.  As
> > > > > we commented earlier and also shown empirically, using 100x100
> > > > > submatrices balances well accuracy and scalability. The
> > > > > exchangeability evaluation (whose purpose is to demonstrate that
> > > > > CNNSAT can identify semantically equivalent formulas) also shows a
> > > > > small (i.e., 0.35%) difference in prediction results.
> > > > >
> > > > > We sincerely hope that these help clarify your questions and improve
> > > > > your evaluation of the work. If you have further questions, we are
> > > > > happy to answer them promptly.

---

> > > > > > ### Comment · AnonReviewer1 · 2018-12-10
> > > > > > **Re: Re 'update'**
> > > > > >
> > > > > > Thanks for the reply.
> > > > > >
> > > > > > > Our wording may be unclear, which we will state more accurately: (1) CNNSAT's focus is on predicting SAT/UNSAT, not solving, and (2) it achieves very accurate and fast SAT/UNSAT prediction.
> > > > > >
> > > > > > -> Hmm, this sounds very different than what the authors said on November 28 in their comment Re Re Re "Comments about Experiments"; here is a quote from then:
> > > > > >
> > > > > > "
> > > > > > > The key message of the paper is that CNNSAT is in position to outperform the state-of-the-art solvers on random 3-SAT instances. Right?
> > > > > >
> > > > > > Yes, that is accurate.   Again, we wish to contend that our experiments are already extensive and demonstrate CNNSAT's orders of magnitude performance gains over state-of-the-art solvers.
> > > > > > "
> > > > > >
> > > > > > This is the sort of statement that I'm worried might lead to a backlash from the SAT community. I'm happy to see a much more cautious tone now (the parts about not replacing but complementing, and about how to paraphrase) and encourage the authors to implement this tone throughout the paper.
> > > > > >
> > > > > >
> > > > > > > [Contributions]
> > > > > >
> > > > > > -> True, (2) is also a contribution.
> > > > > >
> > > > > > > [Scalability]
> > > > > >
> > > > > > -> thanks, these numbers look good; please include them in a future version of the paper.
> > > > > >
> > > > > >
> > > > > > > > Finally, here is a conceptual question: if we always split instances into 10000 submatrices (100x100), and for each of these submatrices record the number of positive and negative literals, then couldn't an instance that has 1000 times more variables but the same number of literals end up with the same representation? That seems very suboptimal.
> > > > > >
> > > > > > > Please note that such two instances would have very different initial representations because of their drastically different sizes, C1xV1 vs.  C2xV2 (where C1, C2 the numbers of clauses, and V1, V2 the numbers of variables in the two instances): C1 = C2, but V1 << V2. Thus, CNNSAT works well with similarly sized instances.
> > > > > >
> > > > > > -> The very different initial representations (yet the same final representation) were indeed the point of my criticism.
> > > > > > I don't understand the last comment "Thus, CNNSAT works well with similarly sized instances." I don't see how that follows from the sentence before?
> > > > > >
> > > > > > > CNNSAT's compressed representation can indeed lead to collisions.  As we commented earlier and also shown empirically, using 100x100 submatrices balances well accuracy and scalability.
> > > > > >
> > > > > > -> I'm not sure whether I got across that my comment shows a conceptual failure mode: with 1000x more clauses, but each of them on average with 1000x less literals, the probability of satisfiability should dramatically increase. Yet, with the 100x100 representation this can lead to the same representation. Since this 100x100 representation, along with the use of convolutions, is the paper's main contribution I would encourage the authors to study such questions in more detail (not to improve the results, but to improve the understanding of the representation's limitations).
> > > > > >
> > > > > >
> > > > > > Overall, I thank the authors for their response, and I believe the paper improved through the iterative review process. Nevertheless, I have remaining doubts about the use of convolutions and the 100x100 matrix representation (i.e., about the paper's main contribution). Because of this, I will keep my score at a 5. If others argue for acceptance, because this is a very interesting and relevant topic I would not stand in the way of acceptance, though (of course based on an expectation that the authors implement the changes discussed).

---

### Author Response · Authors · 2018-11-24
**Revision Summary**

We thank the reviewers for their detailed and helpful feedback, which
has guided us in refining our paper and providing the additional
results to address the main review comments, i.e.,

(1) comparison with state-of-the-art SAT solvers such as PicoSAT and MiniSAT,

(2) experiment to take into consideration the solubility phase transition,

(3) exchangeability experiments, and

(4) experiment to understand how well models trained on one domain
transfer to another domain.

We have updated our paper accordingly, and will incorporate all the
additional results and discussions as permitted by the space
constraints.

We believe that we have carefully addressed all the major comments
from the reviewers, and hope the reviewers also find them responsive
and satisfactory. If the reviewers have further comments, we are happy
to hear about them and will respond accordingly. Thank you for your
valuable time and strong effort in reviewing and considering this
submission for ICLR 2019.

---

### Meta-Review · Area_Chair1 · 2018-12-13
**More experimental evidence needed**

**Confidence:** 4
**Recommendation:** Reject

**Metareview:**

The authors provide a convolutional neural network for predicting the satisfiability of SAT instances. The idea is interesting, and the main novelty in the paper is the use of convolutions in the architecture and a procedure to predict a witness when the formula is satisfiable. However, there are concerns about the suitability of convolutions for this problem because of the permutation invariance of SAT. Empirically, the resulting models are accurate (correctly predicting sat/unsat 90-99% of the time) while taking less time than some existing solvers. However, as pointed out by the reviewers, the empirical results are not sufficient to demonstrate the effectiveness of the approach. I want to thank the authors for the great work they did to address the concerns of the reviewers. The paper significantly improved over the reviewing period, and while it is not yet ready for publication, I want to encourage the authors to keep pushing the idea to further and improve the experimental results.